# International prospective observational study investigating the disease course and heterogeneity of paediatric-onset inflammatory bowel disease: the protocol of the PIBD-SETQuality inception cohort study

Martine A Aardoom,[1] Polychronis Kemos,[2] Irma Tindemans,[3] Marina Aloi,[4] Sibylle Koletzko,[5,6] Arie Levine,[7] Dan Turner,[8] Gigi Veereman,[9] Mattias Neyt,[10] Richard K Russell ,[11] Thomas D Walters,[12] Frank M Ruemmele,[13] Janneke N Samsom,[3] Nicholas M Croft,[2] Lissy de Ridder ,[1] PIBD-SETQuality consortium and PIBD-NET

NMC and LdR are joint senior authors.

For numbered affiliations see end of article.

**Correspondence to**
Dr Lissy de Ridder;
L.deRidder@erasmusmc.nl

## ABSTRACT

**Introduction** Patients with paediatric-onset inflammatory bowel disease (PIBD) may develop a complicated disease course, including growth failure, bowel resection at young age and treatment-related adverse events, all of which can have significant and lasting effects on the patient's development and quality of life. Unfortunately, we are still not able to fully explain the heterogeneity between patients and their disease course and predict which patients will respond to certain therapies or are most at risk of developing a more complicated disease course. To investigate this, large prospective studies with long-term follow-up are needed. Currently, no such European or Asian international cohorts exist. In this international cohort, we aim to evaluate disease course and which patients are most at risk of therapy non-response or development of complicated disease based on patient and disease characteristics, immune pathology and environmental and socioeconomic factors.

**Methods and analysis** In this international prospective observational study, which is part of the PIBD Network for Safety, Efficacy, Treatment and Quality improvement of care (PIBD-SETQuality), children diagnosed with inflammatory bowel disease <18 years are included at diagnosis. The follow-up schedule is in line with standard PIBD care and is intended to continue up to 20 years. Patient and disease characteristics, as well as results of investigations, are collected at baseline and during follow-up. In addition, environmental factors are being assessed (eg, parent's smoking behaviour, dietary factors and antibiotic use). In specific centres with the ability to perform extensive immunological analyses, blood samples and intestinal biopsies are being collected and analysed (flow cytometry, plasma proteomics, mRNA expression and immunohistochemistry) in therapy-naïve patients and during follow-up.

**Ethics and dissemination** Medical ethical approval has been obtained prior to patient recruitment for all sites.

## Strengths and limitations of this study

► International prospective observational study with long-term follow-up examining clinical phenotype and biomarkers that may be predictive for complicated disease course, serious adverse events and response (or not) to therapy.

► The ability to perform extensive immunological analysis (eg, flow cytometry, plasma proteomics, messenger RNA expression and immunohistochemistry) of peripheral blood leucocytes and biopsies in therapy-naïve patients to evaluate heterogeneity at diagnosis and to follow-up these immunological parameters during disease course and therapy use.

► The first prospective PIBD cohort that includes patients from several European and some Asian countries, which enables comparison of therapeutic strategies between countries and continents, and outcomes of these strategies.

► Large scale collection of quality of life measures (eg, IMPACT III and EQ-5D questionnaires) to correlate to clinical findings and compare this across countries and ethnicities.

► Different methods between centres for measurement of laboratory parameters that are part of standard clinical care serve as a limitation to this study.

The results will be disseminated through peer-reviewed scientific publications.

**Trial registration number** NCT03571373.

## INTRODUCTION
### Background
Paediatric-onset inflammatory bowel disease (PIBD) is a chronic disease that often leads

to disabling symptoms such as abdominal pain, diarrhoea and rectal bleeding. Inflammatory bowel disease (IBD) comprises Crohn's disease (CD), ulcerative colitis (UC) and IBD unclassified (IBD-U). Although IBD is most frequently known as an adult disease, in 5%–25% of cases, it is diagnosed during childhood or adolescence.[1–3] The incidence of IBD in Western Europe and North America ranges from 1.85 to 23.82 per 100000 for CD and 1.9 to 23.14 per 100000 for UC.[4] Despite the fact that the incidence of IBD varies among different countries, a current concern is that the general trend shows increasing incidence rates over recent decades, especially among the patients <10 years of age.[1 3 5–7] Compared with adult-onset IBD, PIBD reflects a more severe disease.[8–10] Consequences of the disease, such as growth failure and bowel resection at a young age, may have a large impact on the patient's further development and quality of life (QoL). In addition, the early onset of this disease regularly leads to an early use of intensive therapies with a life-long risk of treatment-related adverse events.[8 11–13]

The pathogenesis of PIBD is currently partly explained by a combination of a genetic predisposition, microbial factors and susceptibility of the immune system leading to an aberrant inflammatory immune response.[14–16] Treatment strategies therefore focus on modulating or suppressing the immune response using immunosuppressive drugs or biologicals. Despite the known heterogeneity within paediatric patients with CD and UC at both disease diagnosis and during follow-up, we are still not able to predict which patients are at risk of developing a complicated disease course and which patients will respond to therapy.[17 18] Therefore, the majority of patients with PIBD have been treated following a step-up approach, a strategy in which patients start with a simpler, easily available therapy at the bottom of the therapeutic pyramid and medications at the top are often considered more efficacious but may present greater risk to the patient. This approach may lead to a delay in treatment response and increases the risk of ongoing inflammation risking penetrating and stricturing complications. The ability to predict complicated disease course and response or non-response to therapy would be of immense value. This is essential to develop strategies that balance, on an individual basis, therapeutic effectiveness with risks of treatment.

In studies including adult patients with IBD, several clinical risk factors have been identified for the development of a complicated disease course in CD and the need for colectomy in patients with UC.[19 20] Given the differences in disease phenotype, course of disease and benefits and risks of treatment between children and adults, findings from studies in adult patients with IBD do not directly apply to PIBD. Few studies have assessed risk factors in patients with paediatric-onset CD and found that both stricturing disease behaviour[17 21 22] and older age at diagnosis[23–25] are associated with an increased risk for the need for surgery. In patients with paediatric-onset UC, the Paediatric Ulcerative Colitis Activity Index (PUCAI)

at diagnosis and 3 months after diagnosis is found to be an essential predictor of colectomy.[26 27] Kugathasan *et al* demonstrated that early anti-tumour necrosis factor (TNF) therapy in CD was associated with a decreased risk of penetrating but not stricturing complications.[28]

Due to the small number of available studies comprising a large variety of data on several different outcomes and predictors combined with a mainly retrospective set-up, to date, most findings regarding predictors of disease course in PIBD are inconclusive. The majority of the identified predictors are demographic or associated with the disease phenotype, and studies lack findings on the predictive value of biomarkers. Despite the known role of the immune system in PIBD, no immunological biomarkers have been identified to correlate with the disease phenotype or disease course. Therefore, there is an urgent need to generate a prospective long-term real-world cohort designed to analyse effectiveness and safety signals with the ability to correlate them to individual risk factors in well-phenotyped patients. To address this issue, the Paediatric Inflammatory Bowel Diseases Network for Safety, Efficacy, Treatment and Quality (PIBD-SETQuality) inception cohort was designed. Besides a few prospective PIBD cohorts, established in the USA and Canada, no international European cohorts currently exist to assess this.[28] Due to possible differences in genotype, environmental influences and treatment algorithms, European data are required. With this paper, we aim to inform the IBD research community about the existence of the PIBD-SETQuality inception cohort and provide insight into the establishment of this cohort.

## Objectives

PIBD-SETQuality is an international project with the overall goal to develop and validate a treatment algorithm for PIBD based on high-risk or low-risk predictors for early complicated or relapsing disease. In this inception cohort, predictors of disease course are identified through prospective collection of longitudinal PIBD data in the first cohort that includes patients from several European and some Asian centres. Table 1 depicts the potentially relevant prognostic factors listed by the consortium, which formed the basis of parameters to investigate in this study.

The main objective of this study is to facilitate the discovery of predictors of disease course, treatment response or non-response and severe adverse events in patients with PIBD by:

1. Collecting real-world longitudinal data with preferably a 20-year follow-up period.
2. Collecting biomaterials and linking this to the detailed clinical data.
3. Using standardised questionnaires to assess QoL.

In addition, this cohort enables investigation of PIBD heterogeneity based on immunological biomarkers and racial or environmental factors. Patients of differing ethnicities in the European countries will be included along with patients from some non-European countries,

**Table 1** Potentially relevant prognostic factors in prediction of disease course and therapy responsiveness in PIBD

| Potentially relevant prognostic factors | |
| --- | --- |
| 1 | Family history |
| 2 | Medical history |
| 3 | Ethnicity |
| 4 | Severity of disease at diagnosis |
| 5 | Disease localisation |
| 6 | Course of disease |
| 7 | Level of inflammatory markers |
| 8 | Faecal calprotectin level |
| 9 | Endoscopic findings |
| 10 | Immunological biomarkers |
| 11 | Genetic polymorphisms |
| 12 | Environmental factors |
| 13 | Dietary factors |
| 14 | Health economic status |
| 15 | Psychosocial status |

which can allow comparison of different races in immigrant and non-immigrant subgroups.

## METHODS AND ANALYSIS
### Study design
The PIBD-SETQuality inception cohort is a multicentre prospective observational study in patients with PIBD. Participants are followed from the moment of diagnosis up to 20 years thereafter. During the first year after diagnosis, data collection is performed more frequently and reduced to annual visits after the second year of follow-up (table 2).

At least 535 patients with PIBD need to be included in the inception cohort. However, since this is an observational study, depending on the feasibility, the study will aim to recruit 1000 patients. In specific centres with the capacity to perform immunological analyses, participants are included in a subcohort for additional collection of biological specimens. This allows in-depth characterisation of immunological pathways in 100 patients with CD and 50 patients with UC and provides the opportunity to relate predictive factors to underlying immune dysfunction during follow-up. The inception cohort is part of the PIBD-SETQuality project, which is funded by Horizon 2020.

### Eligibility criteria
Children and adolescents <18 years with a likely or confirmed diagnosis of IBD are eligible. Diagnosis has to be made or confirmed within the first 2 months after inclusion. Diagnosis must be based on history, physical examination, laboratory, endoscopic, radiological and histological features according to the revised Porto criteria.[29] If a diagnosis of IBD is not confirmed after the investigations are complete, the patient will be excluded from follow-up. Other inclusion criteria comprise available data on all diagnostic procedures for inclusion in the database, informed consent of patient and parents

**Table 2** Visit schedule and included activities for PIBD-SETQuality inception cohort

| Visit number | Visit 0 | Visit 1 | Visit 2 | Visit 3 | Visit 4 | Visit 5 | Annual visits | Unscheduled visit |
| --- | --- | --- | --- | --- | --- | --- | --- | --- |
| Time point | Prior to start of therapy | 4 weeks after start of therapy | 3 months after start of therapy | 6 months after start of therapy | 12 months after start of therapy | 18 months after start of therapy | 24, 36 months and so on | |
| **Activity** | | | | | | | | |
| Study explained, informed consent | X | | | | | | | |
| Collection of routine clinical and laboratory data, including faecal calprotectin | X | X | X | X | X | X | X | X |
| Extra blood sample taken at time of routine blood draw (maximum of 20 mL)* | O | O | O | O | O | O | O | O |
| Extra biopsies taken at time of clinically required colonoscopy (maximum of 8)* | O | | | | | | | O |
| Tissue sample in case of indication for surgical resection* | | O | O | O | O | O | O | O |
| Environmental questionnaire | X | | | | | | | |
| IMPACT III and EQ-5D questionnaires | X (both) | | X (EQ-5D) | X (EQ-5D) | X (both) | X (EQ-5D) | X (both) | |
| School attendance and WPAI questionnaires | X (both) | | X (both) | | X (both) | | X (both) | |

*This activity will only be performed in patients included in the subcohort. 'X' is performed in all patients, 'O' is only performed in patients included in the subcohort.
WPAI, Work Productivity and Activity Impairment.

according to the national guidelines, and in case of inclusion in the subcohort for collection of biomaterial, the patient has not started therapy yet at the moment of diagnostic endoscopy. Patients are excluded from this study: (1) when they are on similar treatments as for IBD but for other conditions, defined as the use of any biological, immunosuppressant or systemic corticosteroid; (2) when they are known to have conditions directly affecting IBD (eg, immunodeficiency or major gastrointestinal resections); and (3) in case of inability to read or understand the patient and family information sheets.

## Recruitment and data collection

All eligible patients are asked for informed consent to participate in the inception cohort. Within this study, clinical data and results of questionnaires will be collected in all patients. Biomaterial is collected as part of the subcohort in specific centres with the ability to perform immunological analyses.

## Clinical data

At baseline, data on demographics, family history, diagnosis, disease activity, disease localisation and results of physical examination, endoscopy, radiographic imaging and laboratory results, including faecal calprotectin levels, are collected. Validated scores and classifications such as the weighted Paediatric Crohn's Disease Activity Index (wPCDAI), PUCAI, Simple Endoscopic Score for Crohn's Disease and Ulcerative Colitis Endoscopic Index of Severity are used.[30–33] During follow-up, the clinical disease activity scores, results of additional investigations and detailed treatment information are collected at fixed time points and during hospitalisations.

## Questionnaires

To assess the QoL, the validated IMPACT III questionnaire is being used. This questionnaire is a disease-specific health-related assessment of QoL, divided in the domains emotional functioning, social functioning, body image and well-being.[34 35] Validated EQ-5D questionnaires are being used to assess the health status of the patient (EQ-5D-Y and EQ-5D-Y proxy) and the parents (EQ-5D-5L). The EQ-5D questionnaire consists of five dimensions: mobility, self-care, usual activities, pain/discomfort and anxiety/depression.[36] Parent's work productivity is assessed by the Work Productivity and Activity Impairment questionnaire (WPAI) for caregivers. All validated questionnaires are used in accordance with their respective instructions regarding age limits. If validated translations of the IMPACT III, EQ-5D and WPAI are not available in a certain language, patients in the respective country will not complete these questionnaires. Due to the lack of validated questionnaires assessing the child's school attendance, a non-validated questionnaire is being used to address this subject. The frequency of the assessment of these questionnaires is shown in table 2. At baseline, many environmental factors are being assessed comprising, for example, parent's smoking behaviour, antibiotic use, sun

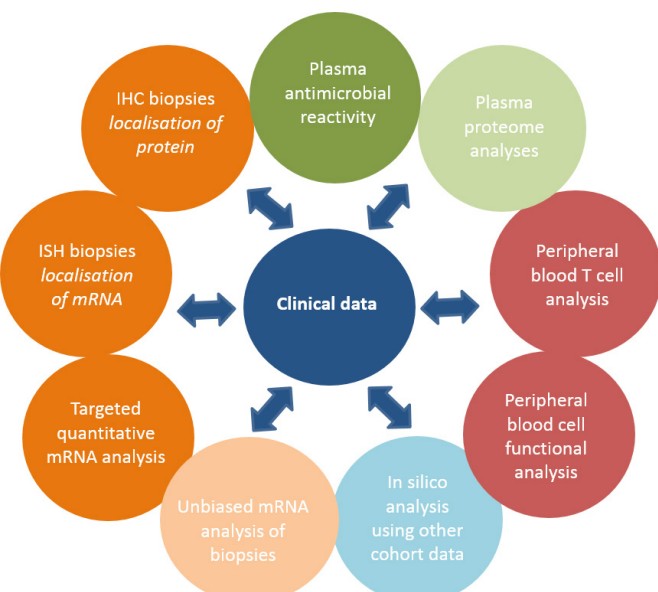

**Figure 1** Different immunological analyses that will be performed. Data will be correlated to the clinical data. IHC, immunohistochemistry; ISH, in situ hybridisation.

exposure, previous enteritis and appendectomy. Assessment of dietary factors is limited to assessing: (1) whether children are following a vegetarian or vegan diet and (2) if they are excluding specific items from their diet.

## Biomaterial

In several centres, biological specimens are collected in addition to the clinical data. When informed consent for the collection of biomaterial is obtained, blood samples and intestinal biopsies are collected prior to treatment initiation and during follow-up visits in concurrence with routine clinical diagnostics. At diagnosis, in all patients, biopsies are collected from affected and non-affected tissue in both ileum and colon according to the standard operating procedure written for this study (online supplementary figure 1). During follow-up endoscopies, collection of ileum biopsies is optional and based on the clinician's decision in patients with UC. Analyses of biopsies include immunohistochemistry, targeted quantitative messenger RNA (mRNA) analyses and unbiased mRNA sequencing analyses. Peripheral blood samples are collected for plasma proteomics analyses, plasma antimicrobial reactivity, in-depth phenotypic and functional leucocyte analyses and screening for genetic polymorphisms (figure 1). Immunological analyses will be related to the previously described clinical data that is being collected. Patients enrolled in the subcohort in whom the diagnosis of IBD is not confirmed after additional investigations will be analysed as non-IBD controls.

## Outcome measures

This study design and statistical analysis plan are based on one primary and several secondary and exploratory outcomes. At baseline, prior to the start of therapy, the clinical characteristics of patients with PIBD will

be reviewed and compared by country, as well as racial background. In patients participating in the subcohort, immunological biomarkers are reviewed in order to classify disease heterogeneity based on underlying immune pathology. Thereto, immunological parameters will be related to clinical and morphological disease features at this time point.

The primary outcome is clinical remission at 1 year, which reflects our aim to study therapy response and non-response. Clinical remission is defined as a wPCDAI <12.5 in patients with CD and a PUCAI <10 in patients with UC or IBD-U. According to the set hypothesis, the disease localisation, extension, behaviour and type of induction therapy are important factors that influence the primary outcome, while age at diagnosis and the initial disease activity score are expected to be important covariates. In addition, biochemical and immunological parameters at baseline will be evaluated to predict clinical remission. In a subanalysis, the effect of early use of immunomodulators and biologicals (less than 3 months from diagnosis) on remission rates will be investigated by comparing groups with early and later use.

One of the secondary outcomes of this study is to assess clinical remission over a period of 3 and 6 months since the start of treatment to evaluate short-term response. In addition, the need for treatment intensification within 1 year will be assessed. The longitudinal follow-up analyses of individual immunological biomarkers will reveal which immune parameters change when patients are in clinical remission. Clinical, biochemical and immunological findings at baseline and 12 weeks after the start of therapy will be assessed for their predictive value on the disease course after 1, 3, 5, 10, 15 and 20 years of follow-up. Lastly, as a longer term secondary outcome, we will assess: (1) moderate or severe disease over the last 6–12 months, (2) the development of complications such as fibrostricturing disease, penetrating disease, active perianal fistula or an abscess, (3) the need for IBD-related luminal surgery and (4) the need for biological use or need for treatment intensification, after 1, 3, 5, 10, 15 and 20 years.

As an exploratory outcome, the initial management of newly diagnosed patients with PIBD will be evaluated. The type of induction therapy and proportion of patients started on early immunomodulators or anti-TNF agents will be assessed and compared by country or region. Several environmental factors will be evaluated to assess their possible role in the heterogeneity of the presenting phenotype as well as disease course. Lastly, longitudinal findings on QoL, health status, parents' work productivity and child's school attendance will be evaluated, compared by country or region and correlated to disease activity and disease course.

## Sample size calculation

To ensure that the study is adequately powered for the primary outcome, we have calculated the minimum required sample size, which we have also adjusted for a 10% expected information loss and multiple comparisons.

To maintain the global significance level alpha at 0.05, for the sample size calculations, we decreased the significance level for the univariate analyses to 0.01 per Bonferroni correction. As we aim to capture predictors with significant influence on clinical remission, we set the effect size of the primary outcome at 25%. In order to detect a 25% or higher difference between the compared groups, as defined by disease characteristics and therapy induction types and considering the variable sampling ratio (2:1 to 1:2 depending on the factor), we will need 203 patients with CD and 214 patients with UC/IBD-U. Considering approximately 40% of PIBD diagnoses comprises UC/IBD-U, we will need 535 patients with PIBD. Subsequently, the CD group will have slightly increased power than the originally planned 80%.

## Data collection and management

Data are collected in REDCap, a secured database, by using online case report forms (CRFs). CRFs are based on the dataset of the Canadian inception cohort by the Canadian Children Inflammatory Bowel Disease Network (CIDsCaNN) and adjusted to the needs of this study. A monitor performs remote (digital) monitoring for each participating centre yearly after inclusion. The coordinating investigator runs consistency checks on a monthly basis and produces queries to be resolved by the local investigators. To secure accurate comparison of biomarkers in tissue specimens obtained from different centres, highly reproducible sample preparation is required across all centres participating in this study. Therefore, all aspects of sample acquisition and all reagents will be strictly regulated, and sample quality will be tightly monitored.

## Analysis and statistical methods

Data analysis will be performed with SPSS V.26 or higher for Windows and R version 3.6 or higher (R Foundation for Statistical Computing, Vienna, Austria). Descriptive statistics will be computed overall and per disease (CD and UC). Phenotypic grouping will be performed according to specific disease characteristics (eg, age of onset, disease localisation and disease behaviour). Next to this, patients will be categorised according to treatment categories. The appropriate descriptive statistics will be used to summarise the demographic characteristics. The primary outcome will be analysed using a 2-proportion Z test for the comparison of the different groups in our sample, using a stricter confidence level of 0.01 per Bonferroni correction. All proportions will be summarised per group, and 95% CIs will be provided. The important factors from the univariate analysis will be used to build a generalised linear model with a logit link function. This will be a multiple analysis of the effects that the selected factors and covariates have on the outcome of clinical remission. The final model will also include interaction terms, if necessary, and will be optimised based on fit diagnostics and residual analysis. The same approach will be used for the secondary outcomes of clinical remission. We will use a mixed-effects linear model to study the effects of the

predictors on the disease activity index over time, taking into account the non-independence between the observations due to the repeated measurements from each patient. For the additional long-term secondary outcomes that have a time-to-event nature, we will use Kaplan-Meier curves to summarise the effects of the categorical factors on the outcome. This will be used to build a Cox regression model with the important factors and covariates that have a significant effect on the time and frequency of the events as defined in the outcomes section.

Similar methods will be used for the analysis of the exploratory outcomes. For these outcomes, propensity scores and multivariate methods are required, in particular for the immunological data, including principal component, factor and cluster analysis. Incidences of serious adverse events (SAE) and the SAE rate per 100 patient-years will be calculated. Missing data analysis will be performed based on the missing at random or missing completely at random mechanisms. Emerging patterns will be thoroughly examined.

## Study status

The first study participant was recruited in 2017. The number of included patients up until February 2020 is 400. Enrolment is expected to be completed by the end of 2021.

## Patient and public involvement

Patients and/or the public were not involved in the initial development, or conduct, or reporting or dissemination plans of this research. However, the French patient charity AFA Crohn, RCH, France, was involved in the final study design and critically reviewed and commented on main aspects of the trial.

## DISCUSSION

The PIBD-SETQuality inception cohort is a unique study, being the first cohort in its size including patients with PIBD from European and Asian countries. The homogeneous collection of data from several different countries in one cohort enables comparison of disease phenotype and treatment paradigms between countries and continents. The hypothesised role of environmental factors in the pathogenesis of PIBD might thus be assessed within this cohort. In addition, being a cohort with real-world data, this study will complement data derived from clinical trials and provide insight in drug use in everyday practice and the related international differences.

The main outcome of this study will be the prediction of a complicated disease course and response or non-response to therapy. One of the important strengths of this study is the collection of biomaterial in therapy-naïve patients. Studies assessing immunological biomarkers in patients with PIBD are scarce and even lacking in therapy-naïve patients. The prospective set-up of this study will reveal whether immunological biomarkers in PIBD change overtime and in response to therapy. In addition,

these biomarkers can be correlated to the clinical data. Besides finding predictive factors of a complicated disease course and response to therapy within this real-world cohort, we will also use this cohort to examine and possibly validate previously reported findings from other studies such as the GROWTH CD study.[37 38]

**Author affiliations**
[1]Department of Paediatric Gastroenterology, Erasmus University Medical Center-Sophia Children's Hospital, Rotterdam, The Netherlands
[2]Centre for Immunobiology, Blizard Institute, Barts and The London School of Medicine and Dentistry, Queen Mary University of London, London, UK
[3]Laboratory of Pediatrics, Division of Gastroenterology and Nutrition, Erasmus University Medical Center-Sophia Children's Hospital, Rotterdam, The Netherlands
[4]Paediatric Gastroenterology and Liver Unit, Department of Paediatrics, Sapienza University of Rome, Rome, Italy
[5]Department of Pediatrics, Dr. von Hauner Children's Hospital, University Hospital, Ludwig Maximilians University Munich, Munich, Germany
[6]Department of Pediatrics, Gastroenterology and Nutrition, School of Medicine Collegium Medicum, University of Warmia and Mazury, Olsztyn, Poland
[7]Paediatric Gastroenterology and Nutrition Unit, Edith Wolfson Medical Center, Tel Aviv University, Holon, Israel
[8]Institute of Paediatric Gastroenterology, Shaare Zedek Medical Center, The Hebrew University of Jerusalem, Jerusalem, Israel
[9]Department of Paediatric Gastroenterology and Nutrition, UZ Brussel, Vrije Universiteit Brussel, Brussels, Belgium
[10]ME-TA Medical Evaluation and Technology Assessment, Merendree, Belgium
[11]Department of Paediatric Gastroenterology, Hepatology and Nutrition, Royal Hospital for Children Glasgow, Glasgow, UK
[12]IBD Centre, Department of Paediatrics, SickKids Hospital, University of Toronto, Toronto, Ontario, Canada
[13]Department of Pediatric Gastroenterology, Université Paris Descartes, Sorbonne Paris Cité, Assistance Publique-Hôpitaux de Paris, Hôpital Necker Enfants Malades, Paris, Île-de-France, France

**Acknowledgements** We would like to thank all patients who participate in this study; everyone who is participating in data entry and all the gastroenterologists who are involved in patient enrolment. Collaborating sites and principal investigators ordered by country: United Kingdom: N Croft, Royal London Children's Hospital, Barts Health NHS Trust London; F Cameron, Alder Hey Children's NHS Foundation, Liverpool; P Henderson, Royal Hospital for Sick Children, Edinburgh; J Ashton, University Hospital Southampton NHS Foundation Trust, Southampton; R Russell, The Royal Hospital for Children Glasgow, Glasgow; M Rafeeq, Birmingham Children's Hospital, Birmingham; N Nedelkopoulou and P Rao, Sheffield Children's Hospital, Sheffield; A Rodrigues, Oxford Children's Hospital, Oxford; J Fell, Chelsea and Westminster Hospital NHS Foundation Trust, London; D Devadason, Nottingham Children's Hospital, Nottingham; J Hart, Royal Devon & Exeter NHS Foundation Trust, Exeter; J Cohen, University College London Hospitals NHS Foundation Trust, London; The Netherlands: L de Ridder, Erasmus Medical Center, Rotterdam; T Hummel, Medisch Spectrum Twente, Enschede; M Wessels, Rijnstate, Arnhem; C van der Feen, Jeroen Bosch Hospital, 's Hertogenbosch; Israel: A Levine, Wolfson Medical Center, Holon; D Turner, Shaare Zedek Medical Center, Jerusalem; A Assa, Schneider Children's Medical Center of Israel, Petach-Tikva Italy: M Aloi, Sapienza University, Rome; France: F Ruemmele, Hôpital Necker Enfants Malades, Paris; Japan: K Arai, National Center for Child Health and Development, Tokyo; Malaysia: WS Lee, University Malay Medical Center, Kuala Lumpur; United Arabic Emirates: C Tzivinikos, Al Jalila Children's Specialty Hospital, Dubai; andGermany: S Koletzko, Dr. von Hauner Children's Hospital, Munich.

**Contributors** MAA prepared the draft manuscript with comments and review from all authors. All authors were involved in the conception, design, planning and drafting of the original research protocol. All authors are involved in the implementation of this study. All authors provided critical review of the manuscript and approved the final version.

**Funding** This project is a Pediatric Inflammatory Bowel Disease Network (PIBD-NET) initiative. It was supported as part of the PIBD Network for Safety, Efficacy, Treatment and Quality improvement of care project funded by the European Commission Horizon 2020, funding source number 668 023. The study sponsor is PIBD-NET.

**Competing interests** MAA received consultation fee and honorarium from Abbvie. SK received consultation fee, research grant or honorarium from Danone, Nestec-Nutrition, Abbvie, Takeda, Celgene, Shire, Pfizer, Biogaia, Janssen, Berlin-Chemie, Mead Johnson, Vifor, Pharmacosmos and ThermoFisher. RKR is supported by an NHS Research Scotland Senior Research Fellowship and has received speakers's fees, travel support, and/or participated in medical board meetings with Nestle, MSD Immunology, AbbVie, Dr Falk, Takeda, Napp, Mead Johnson and Nutricia&4D Pharma. FR has received speaker fees from Shering-Plough, Nestlé, MeadJohnson, Ferring, MSD, Johnson & Johnson, Centocor and AbbVie; has served as a board member for SAC:DEVELOP (Johnson & Johnson), CAPE (AbbVie), LEA (AbbVie); and has been invited to MSD France, Nestlé Nutrition Institute, Nestlé Health Science, Danone, MeadJohnson, Takeda, Celgene, Biogen, Shire, Pfizer and Therakos. NMC (into employer's investigator accounts) received speaker fees, advisory board fees and research funding from Eli-Lilly, Takeda, Abbvie Shire Pfizer and 4D Pharma. LdR had collaborations (such as involved in industry-sponsored studies, investigator-initiated study and consultancy) with Shire, Malinckrodt, Nestlé, Celltrion, Abbvie and Pfizer; and received a grant from ZonMw, ECCO and Pfizer. Remaining authors: no competing interests declared.

**Patient consent for publication** Not required.

**Ethics approval** The study is being conducted according to the principles of the Declaration of Helsinki andso far has been approved by the Medical Ethics Committee Erasmus Medical Centre of Rotterdam, the National Health Service (NHS) Research Authorities (for England and Wales) and the NHS Research Permission Co-ordinating Centre (Scotland) in the United Kingdom,the Ethical Committee of Ludwig-MaximilliansUniversitätMünchen, the Dubai Healthcare City Authority – Regulatory, the University Malaya Medical Centre Medical Ethics Committee, the Ethical Committee of National Center for Child Health and Development in Tokyo, the Committee of Protection of Persons Nord-Ouest I in France, the Institutional Review Board Rabin Medical Center in Israel, the Helsinki Committee of E. Wolfson Medical Center in Israel, ShaareZedek Medical Center Ethics Committee in Israel, the Ethical Committee Messina, the Regional Ethical Committee Delle Marche in Ancona, the Ethical Committee Universita Degli Studi DellaCampania 'Luigi Vanvitelli'—AziendaOspedalierauniversitaria 'Luigi Vanvitelli'—A.O.R.N. 'Ospedali Dei Colli' in Caserta, the Ethical Committee Università Federico II in Naples and the Ethical Committee Unico Regionale in Rome. Handling of patient material complies with the legislation of the relevant country and the European General Data Protection Regulation. In accordance with the H2020 general grant agreement, the dissemination process will ensure open access to the peer-reviewed scientific publications resulting from this project and the related bibliographic meta-data.

**Provenance and peer review** Not commissioned; externally peer reviewed.

**ORCID iDs**
Richard K Russell http://orcid.org/0000-0001-7398-4926
Lissy de Ridder http://orcid.org/0000-0002-6035-1182

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
