## [Reviewer comments · BMJ Open]

ARTICLE DETAILS

TITLE (PROVISIONAL)	An international prospective observational study investigating the disease course and heterogeneity of paediatric-onset inflammatory bowel disease: the protocol of the PIBD-SETQuality inception cohort study
AUTHORS	Aardoom, Martine A; Kemos, Polychronis; Tindemans, Irma; Aloï, Marina; Koletzko, Sibylle; Levine, Arie; Turner, Dan; Veereman, Gigi; Neyt, Mattias; russell, richard; Walters, Thomas D; Ruemmele, Frank; Samsom, Janneke N; Croft, Nicholas; de Ridder, Lissy

VERSION 1 – REVIEW

REVIEWER	Marco Gasparetto The Royal London Children's Hospital Barts Health Trust Department of Paediatric Gastroenterology London, United Kingdom
REVIEW RETURNED	22-Dec-2019

GENERAL COMMENTS	Comments for the Authors: - This work is extremely relevant to the current needs of children and adolescents with IBD. Identification and/or development of prognostic biomarkers is urgently required to improve quality of care by stratifying patients based on their risk for severe disease and by personalising their treatments accordingly.- Title. The title doesn't include information on the specific aim of the study. Given the importance of this work, I would suggest a slight change as follows to grab the reader's eye even more. "A global prospective observational study investigating IBD disease course and its heterogeneity in children and adolescents: the PIBD-SETQuality inception cohort".- Abstract: page 3, line 46 – I would suggest 2-3 examples of the immunological analyses that will be performed and of the biomaterial that will be collected in order to provide the reader with a clearer and more specific understanding of the work planned. Following is a suggestion of examples that could be added to the abstract: "In addition, environmental factors are being assessed (e.g. parent's smoking behaviour, dietary factors, antibiotic use, appendectomy). In specific centres with the ability to perform extensive immunological analyses (e.g. plasma proteome analysis, peripheral blood T cell analysis, mRNA analysis of biopsies), biomaterial (i.e. blood samples and intestinal biopsies) is being collected and analysed in therapy naïve patients at baseline and during follow-up".- Introduction is comprehensive, focused and thorough.
---

	- Article summary (page 5, lines 14 and 27): Clear and focused. Similarly to my comments above, I would provide a few examples of immunological analysis and quality of life measures to inform and guide the reader on the specific analyses planned. Example as follows: “Performance of extensive immunological analysis (e.g. peripheral blood cell functional analysis, target quantitative mRNA analysis) of blood and biopsies in therapy-naïve patients ...” “Large scale-collection of quality of life measures (e.g. IMPACT III and EQ-5D questionnaires) to correlate ... ” - Table 1 on page 9: I would briefly specify how you describe/define disease course, to give the reader a clearer idea of the specific entities analysed. Example as follows: Course of disease (i.e. achievement of remission and/or response to therapy at 3 months, 6 months, 1 year based on wPCDAI, PUCAI; need for treatment escalation within one year; moderate-severe disease during 6-12 months; development of complications such as fibro-stricturing disease, penetrating disease, perianal disease; need for surgery; need for biologic use after 1-3-5-10-15 and 20 years) - Further comments related to Table 1: Ethnicity: this factor may be a confounder in the molecular analyses (e.g. mRNA) because of genetic variations across different ethnic groups and consequent variation in gene expression. Therefore, breaking down the patients’ data by ethnicity is likely to be required. How do you plan to maintain an adequate subpopulation size and statistical power in this respect? Has this been considered in the Power calculation? (i.e. any minimum number of patient per ethnic group required?) Faecal calprotectin is listed in Table 1 but not mentioned in the text, nor in Table 2. Will it be tested and matched to clinical and endoscopic activity scores at different time points to define disease remission or relapse? Diet is in the list of environmental factors that will be analysed: how will you be able to correct for variation across different European and Asian populations? Will the assessment be based on questionnaire(s)? If so, which one(s)? - Eligibility criteria, page 12 line 25 “Children and adolescents < 18 years with a likely or confirmed diagnosis of IBD are eligible.” Biomaterial – page 14, lines 20-24 “Patients enrolled in the subcohort in whom the diagnosis of IBD is not confirmed after additional investigations will be analysed as non-IBD controls”. Please provide more information about the number of control patients you are aiming to. Any power calculations for this? - Eligibility criteria page 12 line 43 – 44: Patients are excluded from this study 1) when they are on similar treatments as for IBD but for other conditions. Please provide some details to this statement. Does this refer to any immune-suppressants and/or corticosteroids and/or biological treatments? - Recruitment and data collection – Biomaterial - page 14, line 3 “In several centres biological specimens are collected in addition to the clinical data ... intestinal biopsies from affected and non-affected tissue are collected prior to treatment initiation ...” – Are
--	--

	biopsies going to be collected from specific gut segments? E.g. terminal ileum, right colon, left colon ... Will it be the same for Crohn's and UC? Please specify how many biopsies will be collected from which gut location. - Outcome measures page 14 In the absence of a unifying score to categorise disease course in IBD, I fully agree with the choice of primary and secondary outcome measures for this study. Irrespective of whether a patient will be managed with a step-up or top-down approach based on the severity of their disease presentation or course, these outcome parameters are likely to capture the main events reflecting disease severity, that will be subsequently correlated to clinical and environmental variables. - Outcome measures - page 15, line 26 - 3) the need for surgery Does this refer to IBD related GI surgery or also to surgery for perianal disease? Please specify. - How many patients have been recruited so far and across how many participating centres? How will you manage missing data/ loss of follow-up? Minor comments: - Page 12, line 31: Please check: "If A diagnosis OF IBD is not confirmed after ..." - Page 14, line 3: Please check: "In several centres biological specimens ARE collected ..."
--	--

REVIEWER	Mikkel Malham Copenhagen University Hospital, Hvidovre, Paediatric Department
REVIEW RETURNED	14-Jan-2020

GENERAL COMMENTS	General: Several papers have been published over the last decade noticing a more severe disease course in paediatric IBD than in adult onset IBD. As the prospective long-term paediatric observational studies are scares, this study is highly relevant and the efforts from the authors in constructing this IBD inception cohort are warmly welcomed. To improve the readability of this paper, however, the authors should revise the language as several sentences are unnecessarily difficult to read due to long sentence structures. Moreover, the main outcome (prediction of complicated disease course and therapy response) should be stated precisely and presented earlier in the protocol. Abstract: PIBD is defined as paediatric IBD. As the study period is 20 years, I think this should be changed to paediatric onset IBD. Do the authors in truth believe that the inclusion of six countries in Asia and Europe is a Global cohort? Summary: 2nd bullet: I am unsure of how to read this point. Is it a strength/limitation that immunological analyses are performed or is it a strength/limitation that you are testing the performance of the immunological analyses. Introduction:
--

	Same objection to the definition of PIBD as in the abstract. The use of the Jess et al reference (number 11) regarding treatment-related adverse effects seems off, as Jess does not report on this subject in the given paper and the main focus is not paediatric onset. Moreover (and importantly), there are several paediatric papers reporting on this specific issue. Objectives: This section is quite vaguely written, and it seems that the authors here state the aims of the study without any objectives. This is a shame as it would improve the oversight of the study for the reader. The authors state that they will assess the impact of diet on disease course, however, I am confused on how the authors will quantify diets. It is stated that they will perform a genetic status. To me that means full GWAS, however, this is not planned, as far as I can understand. Methods and analysis: Table 2: As I understand this is a time-line for the entire study. However, the time-line starts at "time after start of therapy". Should it not be "time from diagnosis"? Eligibility: The authors state that a non-validated questionnaire is utilised to assess school attendance. Would it not be worth the while to validate such a questionnaire before deciding to use it in this very ambitious project? Biomaterial: Are the sites of biopsy harvesting standardized or can the biopsies originate from any part of the GI-tract? This will create a very heterogeneous study material difficult to analyse. mRNA is not defined. Outcome measures: At baseline, clinical characteristics will be compared in patients with active disease. I am confused. Will all patients not be included at diagnosis and will all patients not have active disease at diagnosis? Immunological biomarkers will be evaluated in treatment naïve patients. Are all patients in the study not treatment naïve (with regards to the exclusion criteria)? In line 51: location should be localization. In line 57: earlier should be early. Sample size: The authors state that 535 patients are needed. Previously, however, they stated that at least 500 patients would be included. Moreover, on clinicaltrials.gov (clinicaltrials.gov/ct2/show/NCT03571373) they state that 1000 patients will be included. These numbers should be in agreement.
--	--

REVIEWER	Denise Herzog
-----------------	---------------

	Hôpital cantonal de Fribourg Switzerland
REVIEW RETURNED	16-Jan-2020

GENERAL COMMENTS	With this manuscript the authors publish the protocol of an ongoing global prospective observational study in paediatric IBD (PIBD-SETQuality inception cohort), in order inform interested community members about this study. The most significant new aspect of this protocol is the inclusion of patients before the establishment of the definitive diagnosis. This enables the researchers to examine treatment naive biopsy specimens and blood samples. The second most important aspect is the intended follow-up of twenty years. The collected data will enable the researches to find predictive factors for the different types of disease evolution. The submitted paper is however a summary report of the detailed protocol and therefore some questions remain. Which is the minimum age for inclusion? The questionnaires used are those for school-age children. Are children with very early or early-onset IBD excluded? Are the questionnaires used validated for all European, Arab and Asian languages? As patients are included before endoscopic confirmation of the disease: how many patients will have to be, or have already been excluded after endoscopies because of change of diagnosis? Will these patients serve as controls? How many drop-outs during the 20 years of follow-up do you estimate will occur? In 2019, the rate of patients continuing their participation in the Swiss-IBD cohort study registry initiated in 2006 for adults (in 2008 for children) is 71%. Kugathasan S, et al., Lancet 2017;389:1710-18 planned to enroll 1100 pediatric patients to gain sufficient power for the identification of ten risk factors for B2 and B3 outcomes. Furthermore, they anticipated a 9% drop out over 3 years. In this protocol the power calculation is based on the primary outcome, i.e. clinical remission at one year. Are 535 patients really enough, considering that, after 20 years you also want to find predictive factors for complicated disease evolution? Why do you not take the occasion to collect stool and/or saliva specimens at different time points for microbiological analyses? Why do you plan to stop the follow-up after 20 years? A lifelong registry of IBD patients would be of great value.
--

VERSION 1 – AUTHOR RESPONSE

Reviewers' Comments to Author:

Reviewer: 1

Reviewer Name: Marco Gasparetto

Institution and Country: The Royal London Children's Hospital, Barts Health Trust, Department of Paediatric Gastroenterology, London, United Kingdom Please state any competing interests or state 'None declared': None declared.

I notice that two co-authors are affiliated to the University linked to the Trust I am currently based at. One of these two Authors also practises as a clinician in the same Unit where I am based. Nevertheless, this has not influenced my assessment of the work to any extent as I have never been directly or indirectly involved in this project so far. Therefore, there are no competing interests.

Please leave your comments for the authors below Comments for the Authors:

- 1) This work is extremely relevant to the current needs of children and adolescents with IBD. Identification and/or development of prognostic biomarkers is urgently required to improve quality of care by stratifying patients based on their risk for severe disease and by personalising their treatments accordingly.

We thank the reviewer for his appreciation of this study and the thorough review of this manuscript.

- 2) Title. The title doesn't include information on the specific aim of the study. Given the importance of this work, I would suggest a slight change as follows to grab the reader's eye even more. "A global prospective observational study investigating IBD disease course and its heterogeneity in children and adolescents: the PIBD-SETQuality inception cohort".

We thank the reviewer for his suggestion. We revised the title of the manuscript accordingly, while also taking into account the suggestions from other reviewers and the editor.

- 3) Abstract: page 3, line 46 – I would suggest 2-3 examples of the immunological analyses that will be performed and of the biomaterial that will be collected in order to provide the reader with a clearer and more specific understanding of the work planned. Following is a suggestion of examples that could be added to the abstract: "In addition, environmental factors are being assessed (e.g. parent's smoking behaviour, dietary factors, antibiotic use, appendectomy). In specific centres with the ability to perform extensive immunological analyses (e.g. plasma proteome analysis, peripheral blood T cell analysis, mRNA analysis of biopsies), biomaterial (i.e. blood samples and intestinal biopsies) is being collected and analysed in therapy naïve patients at baseline and during follow-up".

We agree this is an important addition. This was added to the manuscript (page 5, line 80 – 85; version with track changes).

- 4) Introduction is comprehensive, focused and thorough.
Article summary (page 5, lines 14 and 27): Clear and focused.

We thank the reviewer for his compliments.

- 5) Similarly to my comments above, I would provide a few examples of immunological analysis and quality of life measures to inform and guide the reader on the specific analyses planned. Example as follows: "Performance of extensive immunological analysis (e.g. peripheral blood cell functional analysis, target quantitative mRNA analysis) of blood and biopsies in therapy-naïve patients ..."
"Large scale-collection of quality of life measures (e.g. IMPACT III and EQ-5D questionnaires) to correlate ..."

Following this comment we adjusted lines 80 – 85 on page 5 of the manuscript (version with track changes).

- 6) Table 1 on page 9: I would briefly specify how you describe/define disease course, to give the reader a clearer idea of the specific entities analysed. Example as follows:
Course of disease (i.e. achievement of remission and/or response to therapy at 3 months, 6 months, 1 year based on wPCDAI, PUCAI; need for treatment escalation within one year; moderate-severe disease during 6-12 months; development of complications such as fibro-stricturing disease, penetrating disease, perianal disease; need for surgery; need for biologic use after 1-3-5-10-15 and 20 years).

In Table 1 the potentially relevant prognostic factors listed by the consortium are depicted, which formed the basis of parameters to investigate in this study. The specific outcomes and

their definitions are therefore not mentioned in this table. As this may have not been clear enough in the manuscript, we added a sentence in line 180-210 (version with track changes).

7) Further comments related to Table 1:

- a. Ethnicity: this factor may be a confounder in the molecular analyses (e.g. mRNA) because of genetic variations across different ethnic groups and consequent variation in gene expression. Therefore, breaking down the patients' data by ethnicity is likely to be required. How do you plan to maintain an adequate subpopulation size and statistical power in this respect? Has this been considered in the Power calculation? (i.e. any minimum number of patient per ethnic group required?)

We thank the reviewer for pointing this out. In this cohort the overall role of ethnicity in modulating immune responses cannot be addressed. Genetic analyses will be performed to determine whether particular patterns of immune responses associate with known genetic risk factors i.e. SNPs, which can vary depending on ethnicity.

- b. Faecal calprotectin is listed in Table 1 but not mentioned in the text, nor in Table 2. Will it be tested and matched to clinical and endoscopic activity scores at different time points to define disease remission or relapse?

If available, faecal calprotectin levels are recorded in our database at every visit. The faecal calprotectin levels will indeed be transformed to a categorical parameter and used to evaluate disease remission or relapse. Following this comment we added faecal calprotectin levels in Table 2 and line 258 on page 15 (version with track changes).

- c. Diet is in the list of environmental factors that will be analysed: how will you be able to correct for variation across different European and Asian populations? Will the assessment be based on questionnaire(s)? If so, which one(s)?

Thank you for this valid remark. When setting up this study we decided that diet questionnaires or diaries would be too time consuming and too complex in this patient group that is already completing several questionnaires. We therefore decided to ask only two questions; 1) whether they are following a vegetarian or vegan diet, and 2) if they are excluding specific items from their diet and if so, for what reason. We clarified this in the revised manuscript in line 277-282 (version with track changes).

- 8) Eligibility criteria, page 12 line 25 "Children and adolescents < 18 years with a likely or confirmed diagnosis of IBD are eligible." Biomaterial – page 14, lines 20-24 "Patients enrolled in the subcohort in whom the diagnosis of IBD is not confirmed after additional investigations will be analysed as non-IBD controls".

Please provide more information about the number of control patients you are aiming to. Any power calculations for this?

Current data show that approximately 17% of the inclusions in the subcohort are non-IBD controls. As this is a substantial number, we anticipate having sufficient power to detect predominant immune signatures in our patient cohort.

- 9) Eligibility criteria page 12 line 43 – 44: Patients are excluded from this study 1) when they are on similar treatments as for IBD but for other conditions.

Please provide some details to this statement. Does this refer to any immune-suppressants and/or corticosteroids and/or biological treatments?

We thank the reviewer for pointing this out. A definition was added to the manuscript (lines 243-247 version with track changes).

- 10) Recruitment and data collection – Biomaterial - page 14, line 3

"In several centres biological specimens are collected in addition to the clinical data ..."

intestinal biopsies from affected and non-affected tissue are collected prior to treatment initiation ...” – Are biopsies going to be collected from specific gut segments? E.g. terminal ileum, right colon, left colon ... Will it be the same for Crohn’s and UC? Please specify how many biopsies will be collected from which gut location.

Following this comment we adjusted lines 290-294 on page 16 (version with track changes) and added Supplemental Figure 1 that depicts the flow chart for collection of biopsies during the endoscopy.

11) Outcome measures page 14

In the absence of a unifying score to categorise disease course in IBD, I fully agree with the choice of primary and secondary outcome measures for this study. Irrespective of whether a patient will be managed with a step-up or top-down approach based on the severity of their disease presentation or course, these outcome parameters are likely to capture the main events reflecting disease severity, that will be subsequently correlated to clinical and environmental variables.

We thank the reviewer for this feedback.

12) Outcome measures - page 15, line 26 - 3) the need for surgery

Does this refer to IBD related GI surgery or also to surgery for perianal disease? Please specify.

Both luminal surgery and perianal surgery are recorded in this study. However, for this outcome we will focus on luminal IBD-related surgery as perianal disease is captured under 2) in the list of longer-term secondary outcomes. Changes were made in the manuscript to clarify this (line 328).

13) How many patients have been recruited so far and across how many participating centres? How will you manage missing data/ loss of follow-up?

At this moment 400 PIBD patients have been included in the study in twenty different centers. We added sentence to the manuscript on the current status of the recruitment. To reduce the number of missing fields, we have implemented a live system that notifies the sites about missing data and discrepancies. In addition, the data management of the study shares data completion updates with all participating centers on a weekly basis. Due to the large number of participants and to avoid inflation of bias, we are not planning to use any imputation methods. However, we will be focusing on missing at random (MAR) or missing completely at random (MCAR) mechanisms for missing data analysis and thoroughly examine any emerging patterns. Since this is a longitudinal study with repeated measures, we will be using mixed effects models, which are particularly robust to missing visits (lost follow-ups). Regarding the time-to-event analysis, lost to follow-up cases will be censored to the last known visit which makes these subjects valuable for the analysis that will help us maintain the sensitivity for a Kaplan-Meier curve analysis. We added information in line 388-390 (version with track changes) to clarify the missing data handling.

14) Minor comments:

Page 12, line 31: Please check: “If A diagnosis OF IBD is not confirmed after ...”

Page 14, line 3: Please check: “In several centres biological specimens ARE collected ...”

Thank you for pointing this out. We corrected the mistakes in the revised manuscript.

Reviewer: 2

Reviewer Name: Mikkel Malham

Institution and Country: Copenhagen University Hospital, Hvidovre. Denmark.

Please state any competing interests or state ‘None declared’: None declared

General:

Several papers have been published over the last decade noticing a more severe disease course in paediatric IBD than in adult onset IBD. As the prospective long-term paediatric observational studies are scarce, this study is highly relevant and the efforts from the authors in constructing this IBD inception cohort are warmly welcomed. To improve the readability of this paper, however, the authors should revise the language as several sentences are unnecessarily difficult to read due to long sentence structures. Moreover, the main outcome (prediction of complicated disease course and therapy response) should be stated precisely and presented earlier in the protocol.

1) Abstract:

- a. PIBD is defined as paediatric IBD. As the study period is 20 years, I think this should be changed to paediatric onset IBD.
- b. Do the authors in truth believe that the inclusion of six countries in Asia and Europe is a Global cohort?

We thank the reviewer for pointing this out and made changes in the manuscript accordingly.

- 2) Summary: 2nd bullet: I am unsure of how to read this point. Is it a strength/limitation that immunological analyses are performed or is it a strength/limitation that you are testing the performance of the immunological analyses.

The most important strength here is that we obtain extensive immunological data in therapy naïve patients. Subsequently, we are able to follow-up these immunological parameters during the disease course and therapy use. We made changes in the summary section to clarify this.

3) Introduction:

- a. Same objection to the definition of PIBD as in the abstract.
- b. The use of the Jess et al reference (number 11) regarding treatment-related adverse effects seems off, as Jess does not report on this subject in the given paper and the main focus is not paediatric onset. Moreover (and importantly), there are several paediatric papers reporting on this specific issue.

We thank the reviewer for pointing this out. The adjustments were made in the introduction, including the insertion of other references. The reference of Jess et al. was removed and references to paediatric studies that describe disease course and the impact of IBD on children's life were added (reference 8, 11-13).

4) Objectives:

- a. This section is quite vaguely written, and it seems that the authors here state the aims of the study without any objectives. This is a shame as it would improve the oversight of the study for the reader.

We thank the reviewer for pointing this out. We rewrote this section to clarify our objectives.

- b. The authors state that they will assess the impact of diet on disease course, however, I am confused on how the authors will quantify diets.

As this suggestion is based on Table 1 in the manuscript, we clarified the role of this table in the manuscript (line 180-186, version with track changes). In Table 1 the potentially relevant prognostic factors listed by the consortium are depicted, which formed the basis of parameters to investigate in this study. When setting up this study we decided that diet questionnaires or diaries would be too time consuming and too complex in this patient group that is already completing several questionnaires. We therefore decided to ask only two questions; 1) whether they are following a vegetarian or vegan diet, and 2) if they are excluding specific items from their diet and if so, for what reason. We clarified this in the revised manuscript in line 278-282 (version with track changes).

- c. It is stated that they will perform a genetic status. To me that means full GWAS, however, this is not planned, as far as I can understand.

We thank the reviewer for pointing this out. Performing a GWAS in this study was not feasible. Genetic analyses will be performed to determine whether particular patterns of immune responses associate with known genetic risk factors i.e. SNPs. We clarified this in Table 1 so this is in line with the rest of the manuscript.

5) Methods and analysis:

- a. Table 2: As I understand this is a time-line for the entire study. However, the time-line starts at "time after start of therapy". Should it not be "time from diagnosis"?

Following this comment we adjusted the headings in Table 2.

- b. Eligibility: The authors state that a non-validated questionnaire is utilised to assess school attendance. Would it not be worth the while to validate such a questionnaire before deciding to use it in this very ambitious project?

We thank the reviewer for this comment, which is much appreciated. Since it is time consuming to validate such questionnaire and there was (and still is) a clear need for real life data from a paediatric-onset IBD cohort it was decided to, for now, settle with the non-validated questionnaire.

- c. Biomaterial: Are the sites of biopsy harvesting standardized or can the biopsies originate from any part of the GI-tract? This will create a very heterogeneous study material difficult to analyse.

The biopsies are collected following a standardized approach. If possible, biopsies from inflamed and non-inflamed tissue are collected from both the ileum and colon. In case the colon is completely affected or unaffected, biopsies should be taken from the transverse colon. Following this comment we adjusted lines 286-294 on page 16 (version with track changes) and added a supplemental figure that depicts the standardized approach for collection of biopsies during the endoscopy in a flow chart.

- d. mRNA is not defined.

Thank you for pointing this out. We added a definition.

- e. Outcome measures: At baseline, clinical characteristics will be compared in patients with active disease. I am confused. Will all patients not be included at diagnosis and will all patients not have active disease at diagnosis?

We agree this is confusing. We removed "in children with active disease" as all PIBD patients will be included to evaluate this.

- f. Immunological biomarkers will be evaluated in treatment naïve patients. Are all patients in the study not treatment naïve (with regards to the exclusion criteria)?

We thank the reviewer for pointing this out. All patients in the study are indeed treatment naïve. We adjusted the manuscript to clarify this (lines 304-309, version with track changes).

- g. In line 51: location should be localization.

- h. In line 57: earlier should be early.

Thank you for pointing this out. We corrected the mistakes indicated at point g. and h. in the revised manuscript.

- i. Sample size: The authors state that 535 patients are needed. Previously, however, they stated that at least 500 patients would be included. Moreover, on

clinicaltrials.gov (clinicaltrials.gov/ct2/show/NCT03571373) they state that 1000 patients will be included. These numbers should be in agreement.

Based on this comment we corrected the incorrect numbers in the manuscript. According to our formal sample size calculation for the specified primary outcome, which takes into account the UC: CD ratio and drop-outs, we need at least 535 patients. However, this calculation is based on identifying differences in risk factors of 25% or higher and therefore a larger sample size is very welcome. By continuing with recruitment in this observational study up to 1000 patients, we increase our sensitivity and statistical power, also for our secondary outcomes.

Reviewer: 3

Reviewer Name: Denise Herzog

Institution and Country: Hôpital cantonal de Fribourg, Switzerland Please state any competing interests or state 'None declared': none

With this manuscript the authors publish the protocol of an ongoing global prospective observational study in paediatric IBD (PIBD-SETQuality inception cohort), in order inform interested community members about this study. The most significant new aspect of this protocol is the inclusion of patients before the establishment of the definitive diagnosis. This enables the researchers to examine treatment naive biopsy specimens and blood samples. The second most important aspect is the intended follow-up of twenty years. The collected data will enable the researches to find predictive factors for the different types of disease evolution. The submitted paper is however a summary report of the detailed protocol and therefore some questions remain.

- 1) Which is the minimum age for inclusion? The questionnaires used are those for school-age children. Are children with very early or early-onset IBD excluded?

We thank the reviewer for this question. Children aged 0-18 can be included in the study but questionnaires are only obtained from children in accordance with the instruction guidelines of the questionnaire. Which means >9 years of age for the IMPACT questionnaire and >7 years of age for the EQ-5D questionnaire. A sentence was added to the manuscript to clarify (line 268-278, version with track changes).

- 2) Are the questionnaires used validated for all European, Arab and Asian languages?

For the majority of countries a validated translation is available for the IMPACT III questionnaire and EQ-5D. If available, these questionnaires are provided by us to the investigator. In case a validated translation is not available, this questionnaire will not be completed in these countries. However, all the clinical data will be collected as not all our research questions require these questionnaires. We added one sentence in the manuscript to clarify (line 268-278, version with track changes).

- 3) As patients are included before endoscopic confirmation of the disease: how many patients will have to be, or have already been excluded after endoscopies because of change of diagnosis? Will these patients serve as controls?

Patients that are included in the subcohort are indeed analysed as non-IBD controls. Patients that are not included in the subcohort are included around the moment of endoscopy. In case the diagnosis is not confirmed, they are excluded from the study.

- 4) How many drop-outs during the 20 years of follow-up do you estimate will occur? In 2019, the rate of patients continuing their participation in the Swiss-IBD cohort study registry initiated in 2006 for adults (in 2008 for children) is 71%. Kugathasan S, et al., Lancet 2017;389:1710-18 planned to enroll 1100 pediatric patients to gain sufficient power for the identification of ten risk factors for B2 and B3 outcomes. Furthermore, they anticipated a 9% drop out over 3 years. In this protocol the power calculation is based on the primary outcome, i.e. clinical remission at one year. Are 535 patients really enough, considering that, after 20 years you also want to find predictive factors for complicated disease evolution?

In this study we aim for a 20-year follow-up. However, we do realize that this kind of long-term follow-up may result in a relatively high number of drop-outs. The sample size calculation is based on the primary outcome of this study and results in a minimal number of 535 included PIBD patients. This calculation took into account the UC: CD ratio and drop-outs. However, this calculation is based on identifying differences in risk factors of 25% or higher and therefore a larger sample size is very welcome. By continuing with recruitment in this observational study up to 1000 patients, we increase our sensitivity and statistical power, also for our secondary outcomes. The sample size calculation depends on many key settings (type of variables, variance, sensitivity etc.) and therefore a comparison with the Swiss-IBD cohort study is difficult. The sample size needed to evaluate predictive factors of complicated disease evolution is dependent on the percentage of difference we want to observe. For the primary outcome, this was set at 25%, but for the secondary outcomes a lower difference can be used. Our calculations confirm that we have acceptable power to capture significant effects for the primary endpoint as explained in the protocol. Also, it is very likely that we will exceed the set number of 535 PIBD patients as the current recruitment projections suggest that we will have recruited 550 patients by the end of November 2020. We will aim to extend this observational study in order to obtain sufficient power for all secondary outcomes.

- 5) Why do you not take the occasion to collect stool and/or saliva specimens at different time points for microbiological analyses?

We thank the reviewer for this suggestion. We did consider stool collection prior to the start of the study but it was unfortunately not feasible to coordinate and perform this according to high quality standards. Microbiological analysis requires very specific and controlled collection procedures. This would also lead to a significant rise in costs. We could not fulfil these requirements and therefore decided to not perform microbiological analyses.

- 6) Why do you plan to stop the follow-up after 20 years? A lifelong registry of IBD patients would be of great value.

Thank you for this important comment. Indeed, this would be extremely informative and helpful. Since it already is very challenging to strive for a follow-up of 20 years we will do our ultimate best to achieve this. In case we are successful in doing so and will be able to achieve support for doing so, we might even extend further in future.

VERSION 2 – REVIEW

REVIEWER	Gasparetto, Marco Cambridge University Hospitals NHS FT, Paediatric Gastroenterology, Hepatology and Nutrition
REVIEW RETURNED	21-Mar-2020

GENERAL COMMENTS	As I pointed out on my first revision of this paper, identifying and/or developing prognostic biomarkers in IBD is pivotal to the future care of children and adolescents affected by this condition. There is an urgent need for prognostic biomarkers that will allow an improvement of our quality of care by stratifying patients based on their risk for severe disease and by personalising their treatments accordingly. For the above reasons, I fully support the realisation and the publication of this work. I had previously suggested a series of amendments that have now been implemented. I am therefore happy for this paper to be published in its current form.
---

REVIEWER	Mikkel Malham The Paediatric Department, Copenhagen University Hospital, Hvidovre. Denmark
REVIEW RETURNED	24-Mar-2020

GENERAL COMMENTS	Very nice work. All my concerns ha
------------------------------------

REVIEWER	Denise Herzog Cantonal Hospital of Fribourg, Switzerland
REVIEW RETURNED	25-Mar-2020

GENERAL COMMENTS	Thanks for the opportunity to review this important study protocol. The protocol is now very clear. I apologize for not noticing during the first review: Paragraph “Strengths of the study” line 92: The first prospective PIBD cohort that includes patients from several European and some Asian countries” Later it becomes eurocentric: Paragraph “Objectives”, page 10, line 162: In this inception cohort, predictors of disease course are identified through prospective collection of longitudinal PIBD data in the first European cohort.
--

VERSION 2 – AUTHOR RESPONSE

Reviewer 3:

Reviewer Name: Denise Herzog

Institution and Country: Cantonal Hospital of Fribourg, Switzerland Please state any competing interests or state ‘None declared’: no competing interest

1. *Thanks for the opportunity to review this important study protocol. The protocol is now very clear. I apologize for not noticing during the first review:
Paragraph “Strengths of the study” line 92: The first prospective PIBD cohort that includes patients from several *European and some Asian* countries”*

Later it becomes eurocentric:

*Paragraph “Objectives”, page 10, line 162: In this inception cohort, predictors of disease course are identified through prospective collection of longitudinal PIBD data in the *first European* cohort.*

Reply: We would like to thank the reviewer for the compliment and thorough review of our manuscript. Thank you for discovering this inconsistency. In the revised manuscript we changed the sentence in the “Objectives” paragraph (line 162-164). The wording is now similar to the sentence in the paragraph “Strengths of the study”.